# Mental health issues and the association of mental health literacy among adolescents in urban Ethiopia

Hailemariam Mamo Hassen[1]*, Manas Ranjan Behera[2], Deepanjali Behera[2], Ranjit Kumar Dehury[3]

1 Department of Public Health, College of Medicine and Health Science, Dire Dawa University, Dire Dawa, Ethiopia, 2 School of Public Health, Kalinga Institute of Industrial Technology (KIIT) Deemed to be University, Bhubaneswar, India, 3 School of Management Studies, University of Hyderabad, Hyderabad, Telangana, India

* hailemariammamo5@gmail.com

## Abstract

**Data Availability Statement:** All relevant data are within the paper and its Supporting Information files.

### Background

Epidemiological evidence about the prevalence of adolescent mental health issues and their association with mental health literacy is crucial for sustained mental health promotion strategies. Adolescence is a critical life stage for mental health promotion. However, evidence is not available among Ethiopian school adolescents. Hence, the present study examined the prevalence of adolescents' mental health issues and their correlation with mental health literacy.

### Materials and methods

A cross-sectional study was conducted among adolescents (grades 5–12) in Dire Dawa city, Eastern Ethiopia using multistage random sampling. Data was collected using the Strength and Difficulty Questionnaire, WHO-5 well-being index, and mental health literacy questionnaire. SPSS version 25 was used for the descriptive, Chi-square, binary logistic regression, and correlation analyses.

### Results

Between 14.0–24.5% of adolescents had reported mental health problems: internalizing problems (14.9–28.8%), emotional problems (10.4–25.5%), and peer relationship problems (17.8–25.5%). These mental health problems were significantly greater among adolescents who had either themselves or their family members used psychoactive substances (p≤0.05). Females from upper elementary (5–8 grade) and lower secondary (9–10) grade levels had a higher prevalence of mental health problems (AOR: 2.60 (0.95–7.10, p<0.05)). The effect of age, parental education, or employment status was insignificant (p>0.05). The prevalence of depression ranged from 18.0–25.5%. Mental health literacy was negatively correlated with total difficulties scores and positively associated with mental well-being scores (p<0.05).

**Funding:** The authors received no specific funding for this work.

**Competing interests:** The authors have declared that no competing interests exist.

## Conclusion

The prevalence of adolescents' mental health problems was higher. It implied that promoting mental health literacy could enhance adolescents' positive mental health. Intervention programs should prioritize vulnerable groups and individuals reporting symptoms of mental health difficulties. Future studies should involve qualitative studies and consider effect of other determinants.

## 1. Introduction

Globally, mental health problems disproportionately affect adolescents in low-income countries and they remain the leading causes of disease burden among children and young people [1]. Knowing about these mental health conditions, their symptoms, and treatments termed as mental health literacy is essential for everyone but is crucial for adolescents [2]. Mental health literacy is defined as understanding and applying information about mental health [2]. In this regard, much has been known about the mental health literacy of adolescents from high-income countries [3]. However, it is still a relatively new concept in Africa [2]. Additionally, studies revealed that access to mental health information or services in Africa is lower due to poverty, lack of education, and cultural stigma around mental illness [1]. Nevertheless, limited evidence in Africa, including Ethiopia, has indicated a growing awareness about the importance of mental health literacy. Despite the challenges associated with poverty, cultural stigma, lack of access to health facilities, and lower education coverage, countries are working to improve mental health literacy [2].

The Ethiopian Ministry of Health has developed a mental health policy focusing on primary mental healthcare and promoting positive mental health, including a mental health awareness campaign [4]. The present study area is Dire Dawa, one of Ethiopian provinces known for its ethnic and cultural language diversity. A recent study revealed that adolescent mental health literacy from Dire Dawa city was lower, consistent with similar studies reports and showing variability across socio-demographic factors [5].

Epidemiological evidence about the prevalence of mental health problems and their association with mental health literacy is crucial for sustained mental health promotion strategies, as adolescence is a critical life stage for mental health promotion [6–10]. However, such evidence is rarely available among Ethiopian school adolescents. Adolescent mental health has piqued the interest of public health professionals due to the unique nature of adolescence [11]. Therefore, adolescence remains a critical life stage for mental health promotion [11]. A popular saying goes, "there is no health without mental health" [12]. With multifaceted burdens, mental health is integrally linked to physical health and well-being [13]. The typical adolescent mental health problems include but are not limited to substance use/abuse or disorders, depression/mood disorders, anxiety, schizophrenia, stress/neurosis, behavioral disorders, postpartum psychosis, and posttraumatic stress disorder (PTSD) [14, 15]. Several studies showed that these mental health problems in adulthood occur during adolescence [14, 15].

For multiple reasons, adolescents are among the most important groups for mental health studies. For example, adolescence is key to physical, emotional, and cognitive progress [11, 16, 17]. Fundamentally, mental, physical, and emotional developmental processes evolve during adolescence [18]. Also public health perspectives emphasizing mental health information during early childhood, [18] would help in later life in managing and preventing mental health problems [18].

Mental health literacy is essential to preventing mental illnesses, consisting of knowledge about mental health problems and recognizing illnesses with their symptoms [1, 3]. Studying the mental health status and its association with the mental health literacy of adolescents is paramount because it helps understand these variables across varied cultural settings [1]. The peak of the onset of mental disorders mainly occurs in the adolescent period and needs urgent attention [19]. In low-resource countries like Ethiopia, risk factors such as cigarette smoking, alcohol consumption, early cohabitation, unhealthy lifestyles etc, are higher among adolescents [20, 21]. leading to various mental health problems [11, 16, 17]. Hence, epidemiological methods like cross-sectional studies provide empirical evidence about the mental health status and factors that contribute to mental health problems [22]. Early understanding of this evidence contribute for making informed and effective interventions to improve adolescents' mental health [23]. As a result, there is an increasing demand for adolescents' mental health and well-being outcome measures to inform the public and policymakers regarding individual or service-level health promotion and therapeutic practices [24]. In this regard, the Strength Difficulties Questionnaire (SDQ) and the Mental Well-being Index (WHO-5) have been essential to meet the criteria of valid and reliable measures.

The assessment of mental health issues using the familiar Diagnostic Standard Manual– 5 (DSM-5) and International Classification of Disease (ICD-11) analysis has been published in previous studies [25, 26]. Such an approach is marked by clustering disorders, namely, externalizing and internalizing problems. Instruments with high validity are routinely used to measure the internalizing groups (i.e., depressive, somatic, and anxiety symptoms) and the externalizing groups (substance use, disruptive, and conduct symptoms). Furthermore, the well-being state was examined using the WHO-5 index, with lower scores indicating depression.

The daily suffering of people with mental illness affects family members, mainly children, and adolescent populations [27]. Globally, about 25% of the total population and 10–20% of children and adolescents have mental disorders, of which half of all these mental health problems begin by the age of fourteen and three-fourths by the age of early twenties [28, 29]. A worldwide-pooled prevalence meta-analysis on mental disorders from all regions of the world addressing 27 countries showed a prevalence of 13.4% ranging from 11.3 to 15.9 (CI: 95%), which indicates mental disorders are affecting a large proportion of the child and adolescent population [30]. According to studies that used the strengths and difficulties questionnaire (SDQ), around 10.5% (5.8–15%) of adolescents in developing countries had mental health concerns [31].

The prevalence of mental health problems among adolescents in Africa including Ethiopia, was reportedly higher [32–36]. For instance, a meta-analysis study on adolescents in sub-Saharan Africa, including Ethiopia (N = 9713), showed the proportion of adolescents having psychopathology was 14.3% (13.6%-15.0%), and the prevalence of psychological disorders was 19.8% (18.8%-20.7%) [34]. These mental health problems are exacerbated by modifiable factors such as low mental health service seeking, lower mental health service use, stigma, discrimination, and common human rights abuses. These factors are linked to immediate and intermediate mental health outcomes such as health literacy and behaviour. Mental health literacy is a changeable factor linked to improved psychological outcomes [6–10]. It refers to how well a person understands mental diseases and the associated factors [6–10]. Mental health literacy includes an individual's potential to identify mental distress [9, 37–39]. Some qualitative descriptions revealed that mental health difficulties and mental well-being are associated with mental health literacy, despite the lack of quantitative evidence of the correlation between mental health literacy, mental health status, and mental well-being [30].

There has been a scarcity of available epidemiological evidence about mental health conditions and literacy among Ethiopian school adolescents. There was also a lack of evidence about school adolescents' mental health status and the effect of socio-demographic characteristics in the existing literature. Therefore, the objectives of the study were:

- To assess the prevalence of mental health issues among urban adolescents in Ethiopia

- To evaluate the relationship between mental health literacy and mental health issues and

- To examine the association between socio-demographic characteristics and mental health issues

## 2. Materials and methods

### 2.1. Study design and sampling

The present research was a cross-sectional study among public and private school adolescents in grades 5 through 12 from 11 to 19 years in Dire Dawa, Ethiopia. A combination of multi-stage (schools, classrooms, then individual students) and systematic and random (using the list of the students in fixed intervals of their roll numbers) was used to select study participants from public and private schools. The prevalence of self-reported mental health difficulty was obtained from the pilot study of the same population (p = 0.378) [40], and the margin of sampling error (d = 0.04) was taken to calculate the sample size.

$$n \geq \frac{(Za/2)^2 \text{ x } P(1-P)}{d^2} = \frac{1.96^2 \text{ x } 0.378(1-0.378)}{0.04^2} = 559.673 \approx 560$$

Data was collected from June 2020 to July 2020 using the mental health literacy questionnaire (MHLQ), the Strength and Difficulty Questionnaire (SDQ), and the WHO-5 well-being index after cross-cultural validation in the study settings. Maximum sample size estimation (n = 924) was approached after taking design effect (d = 1.5) and a 10% non-response. Eighty one percent of potential participants (n = 751) filled out the questionnaires out of the 924 potential participants.

### 2.2. Data collection tools

Data collection tools were the Strengths and Difficulties Questionnaire (SDQ), mental health literacy questionnaire (MHLQ), and mental well-being index (WHO-5). The predictor variables mainly focusing on age, grade level, self or any family member experience with psychoactive substance use, parents practising corporal punishment, perceived worry about family problems, parents' education, and job status were collected by using questionnaires. The Strengths and Difficulties Questionnaire (SDQ) is a valid, rapid measure for emotional and behavioral problems worldwide [31, 41–44]. The SDQ is a 25-item self-report questionnaire that assesses five mental health domains: conduct problems, emotional symptoms, peer problems, hyperactivity-inattention, and pro-social behavior [31, 41–44]. It is a 25-item, 3-point Likert scale (0 = not true, 1 = somewhat true, and 2 = certainly true). It measures subscale difficulty problems that are emotional(SDQ3,8,13,16&24), conduct (SDQ5,7,12,18&22), hyperactivity/inattention(SDQ 2,10,15,21&25), peer(SDQ6,11,14,19&23) and pro-social behaviour (SDQ 1,4,9,17&20) problems [45]. The total difficulties score combines conduct, hyperactivity, emotional, and peer relationship behaviour problems. The externalizing problem manifests emotional and peer behavioural relationship problems. Meanwhile, the internalizing problem manifests the conduct and hyperactivity/inattention problems.

Adding up the first four subscales leads to total difficulty scores (the higher the total difficulties score, the more significant the mental health difficulty) [45]. The fifth sub-scale of the SDQ reflects pro-social behaviour (the higher the score, the better the pro-social behaviour) [45]. SDQ is widely used in resource-poor countries to measure children and adolescents' mental and emotional problems. The baseline for cut-off scores, which defined total and sub-scales strength and difficulty scores were from the United Kingdom population norms defined during the instrument development [43, 46]. The self-administered SDQ is a validated and widely used measurement tool for determining mental health problems in children and adolescents [31, 41–44]. It has already been translated into over 60 languages, including Ethiopian Amharic [31, 41–44]. It has reportedly been an effective tool for assessing children's mental health in a recent scoping review in Africa [47].

Mental health literacy was measured using the mental health literacy questionnaire (MHLQ). This MHLQ tool is freely available and validated for adults, [48] and adolescents [7, 39, 49–56] living in low-income countries to measure health literacy scores. In the context of the current investigation, its reliability was evaluated using Cronbach's alpha and found to be 0.834. MHLQ has 33 items with five Likert scale ratings *(1 = strongly disagree, 2 = slightly disagree, 3 = neither agree nor disagree, 4 = slightly agree, 5 = strongly agree)*. The MHLQ consists of 33 questions that measure factors like recognition (10 questions), knowledge (8 questions), attitudes (8 questions), and beliefs (7 questions) of the adolescents on mental health status [6–10]. The range of scores for the mental health literacy tool is 33 to 165. A higher score implies a better level of mental health literacy.

The mental well-being index (WHO-5) is an overarching expression of the quality of the various domains in the life of adolescents subjective to their mental and psychological well-being. Mental well-being was assessed using the WHO's well-being index, which consists of 5 items (WHO-5) [57–62]. It is a Likert-type scale with a five-point scale (5 = all of the time, 4 = most of the time, 3 = more than half the time, 2 = less than half the time, 1 = some of the time, 0 = at no time).

## 2.3. Ethical approval

Ethical approval was obtained from KIIT University, Bhubaneswar, India and Haramaya University, Ethiopia. Written informed consent was obtained from all participants. Written informed consent was obtained from parents of adolescents aged under fifteen years had offered their consent and adolescents of fifteen and above years was factored out following the school counsellors and school principals' adequate explanation about their level of maturity and decisional capacity. As a duly authorized representative, written informed consent was obtained from school principals.

## 2.4. Statistical analysis

SPSS version 25 was used to carry out the statistical analysis. Descriptive analysis was performed to present the socio-demographic characteristics, cut-off score determination, and prevalence of mental health problems. Chi-square test and binary logistic regression analysis were used to estimate the difference and compare the mental health problems prevalence between males and females across some socio-demographic characteristics. Correlation analysis was performed to assess the associations between mental health literacy, strength difficulties questionnaire scale, and subjective mental well-being. Estimates used a confidence interval of 95% (p≤0.05).

The cut-off score for mental health problems(cases) was defined using the original 3 band categorizations (cut-off scores at 80th & 90th) [63–65]. Scores of every item in the scale are

added from 0–10. The added scores (range of 0–40) were calculated by adding scores on the emotion components, conduct problems, hyperactivity, inattention, and peer problem scales [31, 43, 44]. The cut-off scores was defined by percentile into the highest first 10% (abnormal), the next 10% (borderline), and the remaining 80% (normal). The cut-off for mental health problems (caseness) was the highest at first 10%.

## 3. Results

### 3.1. The socio-demographic characteristics of study participants

The socio-demographic characteristics of study participants are presented in **Table 1**. The study included 731 adolescents (366 males, 365 females) aged 11–19 years old, with a mean age of 16.11 years. The majority of participants were from middle adolescent age groups (40.6%), followed by late adolescents (46.1%) and early adolescents (13.3%). Most participants had at least a secondary education level (71.2%), and a majority were not using psychoactive substances (89.7%).

### 3.2. Strength difficulty scores and prevalence of mental health status

The original 3 bands (cut-off scores at 80th & 90th) and the newer 4 band (cut-off scores at 80th, 90th & 95th) categorization were reported just for comparison as reported in **Table 2**. The original 3 band was used to determine and estimate adolescents' mental health issues. The mental health problems expressed with the total difficulties and in subscales, in general, were higher for female adolescents than male adolescents. The prevalence of mental health problems (cases) was presented across gender and age categories.

As indicated in **Fig 1**, the prevalence of mental health total difficulties (14.0–24.5%), internalising (14.9–28.8%), emotional (10.4–25.5%), and peer relationship (17.8–25.5%) problems were higher compared to reports of previous studies. These mental health problems were more prevalent for female and male adolescents compared to other age groups in extent of percentage as explicitly visualized in Fig 1 across the three age groups. Prevalence of depression ranged from 18.0%-25.2%, reportedly higher for female adolescents aged 14–16 years.

### 3.3. Association of mental health status and socio-demographic factors

Statistical difference in prevalence was examined for each banding of subscale using the Chi-square test. The proportion of abnormal females with total difficulties, emotional and internalizing problems was significantly higher (p<0.05) than males. The prevalence of conduct and externalizing problems was highly associated with being male. The association of socio-demographic characteristics with mental health status was analyzed using Chi-square test as reported in **Table 3**.

There was a significant difference in the prevalence of total difficulties, emotional problems and internalizing problems (Table 3). Males and females experienced similar rates of conduct problems, hyperactivity, and externalizing problems. However, males were more likely to report emotional problems ($\chi^2$ = 14.442, p = 0.001), while females were more likely to report internalizing problems ($\chi^2$ = 9.332, p = 0.009). Overall, there was a significant difference in the prevalence of total difficulties between males and females ($\chi^2$ = 6.913, p = 0.042).

A binary logistic regression model analysis showed that the odds of mental health problems were significantly associated with some socio-demographic characteristics (p<0.05); odds were different across these characteristics as reported in **Table 4**. The prevalence of mental health problems was significantly higher for females in upper elementary (AOR: 2.60 (0.95–7.10)) and lower secondary levels (AOR: 2.73 (1.19–6.29)) compared to upper secondary grade

**Table 1. Descriptive statistics of socio-demographic characteristics stratified by gender and mean age.**

| Socio-demographic Characteristics | Male n (%) | Female n(%) | Total n(%) | Age in years Mean ± SD |
|---|---|---|---|---|
| All Participants | 366(50.1) | 365(49.9) | 731(100) | 16.11±2.11 |
| Age group(years) | | | | |
| Early adolescents(11–13) | 50(13.7) | 47(12.9) | 97(13.3) | 12.43±0.67 |
| Middle adolescents(14–16) | 134(36.6) | 163(44.7) | 297(40.6) | 15.15±0.80 |
| Late adolescents(17–19) | 182(49.7) | 155(42.5) | 337(46.1) | 18.01±0.81 |
| Grade level | | | | |
| Upper elementary (Grade 5–8) | 144(39.3) | 157(43.0) | 301(41.2) | 14.38±1.78 |
| Lower Secondary (Grade 9–10) | 145(39.6) | 146(40.0) | 291(39.8) | 16.88±1.30 |
| Upper secondary (Grade 11–12) | 77(21.0) | 62(17.0) | 139(19.0) | 18.24±0.87 |
| Self-experience on psychoactive substances use | | | | |
| No | 318(48.48) | 338(51.52) | 656(89.74) | 16.04±2.11 |
| Yes | 48(64.00) | 27(36.00) | 75(10.26) | 16.76±2.00 |
| Any family experience on psychoactive substances use | | | | |
| No | 253(72.86) | 271(70.44) | 524(77.12) | 15.99±2.14 |
| Yes | 95(27.14) | 113(29.66) | 208(22.88) | 16.64±1.99 |
| Parents practicing corporal punishment | | | | |
| No | 120(48.26) | 134(51.74) | 254(65.70) | 16.08±2.08 |
| Yes | 129(57.33) | 96(42.67) | 225(34.30) | 16.50±1.98 |
| Adolescents report perceived worry about family problem | | | | |
| No | 110(47.83) | 120(52.17) | 230(31.46) | 15.62±2.28 |
| Yes | 256(51.10) | 245(48.90) | 501(68.54) | 16.34±1.99 |
| Mother education level | | | | |
| Non-educated | 233(63.7) | 221(60.5) | 454(62.1) | 16.15±2.16 |
| Elementary Level | 31(8.5) | 34(9.3) | 65(8.9) | 16.78±1.94 |
| Secondary level | 73(19.9) | 86(23.6) | 159(21.8) | 15.96±2.04 |
| College or above | 29(7.9) | 24(6.6) | 53(7.3) | 15.42±1.91 |
| Father education level | | | | |
| Non-educated | 200(54.6) | 198(54.2) | 398(54.4) | 16.19±2.16 |
| Elementary Level | 21(5.7) | 9(2.5) | 30(4.1) | 17.00±1.55 |
| Secondary level | 90(24.6) | 103(28.2) | 193(26.4) | 15.85±2.10 |
| Diploma/Certificate or above | 55(15.0) | 55(15.1) | 110(15.0) | 16.05±2.01 |
| Mother Job | | | | |
| Housewife Or Unemployed | 129(45.9) | 153(53.1) | 282(49.6) | 16.39±2.07 |
| Work in private | 106(37.7) | 91(31.6) | 197(34.6) | 16.13±2.00 |
| Employed | 46(16.4) | 44(15.3) | 90(15.8) | 15.38±2.23 |
| Father Job | | | | |
| Unemployed/Jobless | 13(3.6) | 6(1.6) | 19(2.6) | 17.26±1.45 |
| Work in private | 257(70.2) | 256(70.1) | 513(70.2) | 16.17±2.10 |
| Employed | 96(26.2) | 103(28.2) | 199(27.2) | 15.84±2.15 |

level (p<0.05); but it was not significantly different for males. It was twice higher among males adolescents with a self-psychoactive substance use (AOR: 2.20(1.07–4.52)) and family members experiencing psychoactive substance use (AOR:2.04(1.01–4.09)). Prevalence differences across age, maternal and paternal education, and employment status were insignificant (p>0.05).

**Table 2.  The cut of scores for baseline reference ranges (caseness) for both the original 3 band and the newer 4 band categorizations.**

| Self-completed SDQ | Original 3 band categories(cut-off scores at 80th & 90th) | | | | | | Newer 4 band categorisations (cut-off scores at 80th, 90th & 95th) | | | | | | | |
|---|---|---|---|---|---|---|---|---|---|---|---|---|---|---|
| | Normal | | Borderline | | Abnormal | | Close to average | | Slightly raised slightly lowered | | High (/low) | | Very high (/very low | |
| | Baseline norm | Present study range | Baseline norm | Present study range | Baseline norm | Present study range | Baseline norm | Present study range | Baseline norm | Present study range | Baseline norm | Present study range | Baseline norm | Present study range |
| Total difficulties score | 0–15 | 0–14 | 16–19 | 15–17 | 20–40 | 18–40 | 0–14 | 0–14 | 15–17 | 15–17 | 18–19 | 18–20 | 20–40 | 20–40 |
| Emotional problems score | 0–5 | 0–5 | 6 | 6 | 7–10 | 7–10 | 0–4 | 0–5 | 5 | 6 | 6 | 7 | 7–10 | 8–10 |
| Conduct problems score | 0–3 | 0–3 | 4 | 4 | 5–10 | 5–10 | 0–3 | 0–3 | 4 | 4 | 5 | 5 | 6–10 | 6–10 |
| Hyperactivity score | 0–5 | 0–4 | 6 | 5 | 7–10 | 6–10 | 0–5 | 0–4 | 6 | 5 | 7 | 6 | 8–10 | 7–9 |
| Peer problems score | 0–3 | 0–4 | 4–5 | 5 | 6–10 | 6–10 | 0–2 | 0–4 | 3 | 5 | 4 | 6 | 5–10 | 7–9 |
| Pro-social score | 6–10 | 6–10 | 5 | 5 | 0–4 | 0–4 | 7–10 | 7–10 | 6 | 6 | 5 | 5 | 0–4 | 0–4 |
| Externalising score | 0–5 | 0–5 | 6–10 | 6–7 | 11–20 | 8–20 | 0–5 | 0–5 | 6–10 | 6–7 | 11–12 | 8 | 13–20 | 9–20 |
| Internalising score | 0–4 | 0–8 | 5–8 | 9–10 | 9–20 | 11–20 | 0–4 | 0–8 | 5–8 | 9–10 | 9–10 | 11–12 | 11–20 | 13–20 |

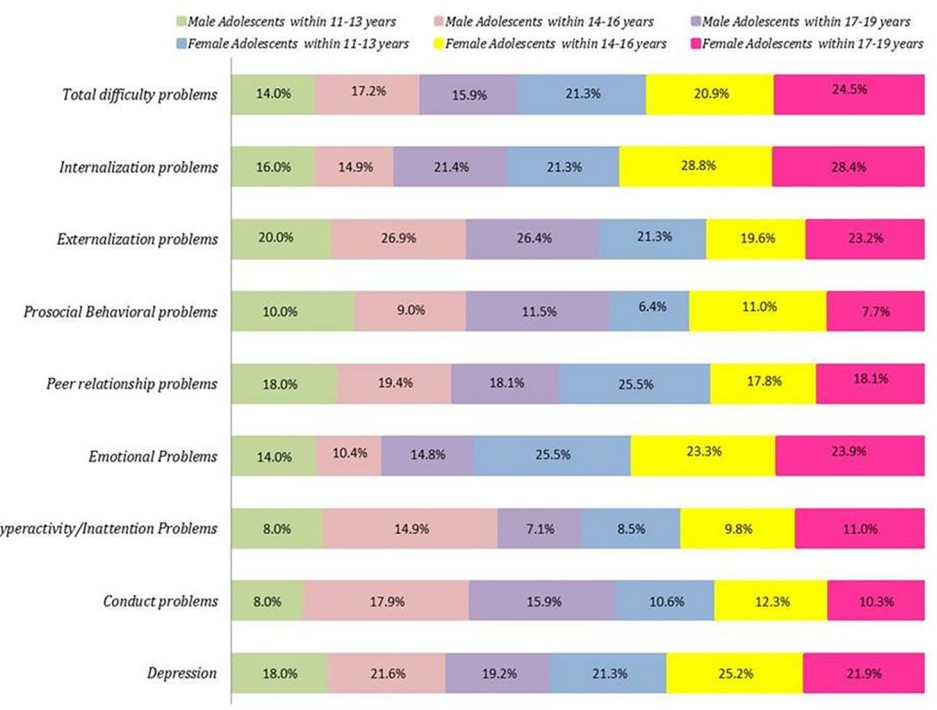

**Fig 1.  Prevalence of mental health problems among school adolescents in Dire Dawa, Ethiopia.**

**Table 3. Chi-square test of mental health problems prevalence as per the original 3 band categorization (cut-off scores at 80th & 90th) between males and females.**

| Subscales and bandings | Males (n = 366) | | Females (n = 365) | | df | $X^2$ | p |
|---|---|---|---|---|---|---|---|
| | n | % | n | % | | | |
| Total difficulties | | | | | 2 | 6.913[a] | 0.042 |
| Normal(0–14) | 307 | 83.9% | 283 | 77.5% | | | |
| Borderline(15–17) | 29 | 7.9% | 48 | 13.2% | | | |
| Abnormal(18–40) | 30 | 8.2% | 34 | 9.3% | | | |
| Emotional problems score | | | | | 2 | 14.442[a] | 0.001 |
| Normal0-5) | 318 | 86.9% | 278 | 76.2% | | | |
| Borderline = 6 | 23 | 6.3% | 36 | 9.9% | | | |
| Abnormal(7–10) | 25 | 6.8% | 51 | 14.0% | | | |
| Conduct problems | | | | | | | |
| Normal(0–3) | 309 | 84.4% | 324 | 88.8% | 2 | 3.674[a] | 0.159 |
| Borderline = 4 | 30 | 8.2% | 18 | 4.9% | | | |
| Abnormal(5–10) | 25.0 | 7.4% | 25.0 | 6.3% | | | |
| Hyperactivity score | | | | | 2 | 0.488[a] | 0.783 |
| Normal (0–4) | 329 | 89.9% | 328 | 89.9% | | | |
| Borderline = 5 | 21 | 5.7% | 18 | 4.9% | | | |
| Abnormal(6–10) | 16 | 4.4% | 19 | 5.2% | | | |
| Peer problems score | | | | | 2 | 0.930[a] | 0.628 |
| Normal(0–4) | 298 | 81.4% | 296 | 81.1% | | | |
| Borderline = 5 | 41 | 11.2% | 36 | 9.9% | | | |
| Abnormal(6–10) | 27 | 7.4% | 33 | 9.0% | | | |
| Internalising score | | | | | 2 | 9.332[a] | 0.009 |
| Normal(0–8) | 299 | 81.7% | 264 | 72.3% | | | |
| Borderline(9–10) | 36 | 9.8% | 50 | 13.7% | | | |
| Abnormal(11–20) | 31 | 8.5% | 51 | 14.0% | | | |
| Externalising score | | | | | 2 | 1.896[a] | 0.387 |
| Normal(0–5) | 272 | 74.3% | 287 | 78.6% | | | |
| Borderline(6–7) | 50 | 13.7% | 41 | 11.2% | | | |
| Abnormal(8–20) | 44 | 12.0% | 37 | 10.1% | | | |
| Pro-social score | | | | | 2 | 0.549[a] | 0.760 |
| Normal(6–10) | 328 | 89.6% | 332 | 91.0% | | | |
| Borderline = 5 | 18 | 4.9% | 14 | 3.8% | | | |
| Abnormal(0–4) | 20 | 5.5% | 19 | 5.2% | | | |

However, the prevalence of mental health problems was not significantly different for males by education level. Prevalence difference was insignificant across the three age groups (11–13, 14–16, and 17–19 years), maternal and paternal education level, and types of jobs or level of employment (p>0.05). The prevalence of mental health problems was twice higher among psychoactive substance users (p<0.05) compared with non-users. Differences in prevalence existed but were insignificant across these three age groups, maternal and paternal education level, and types of jobs or level of employment (p>0.05).

## 3.4. Correlation of mental health literacy with total difficulty scores and well-being index

The bivariate correlations showed that the mental health literacy score was negatively associated with the total and subscale strength difficulties scores (conduct problems, emotional problems, hyperactivity-inattention, and peer problems) as it is shown in **Table 5**.

**Table 4. Association of mental health problems and socio-demographic determinants analyzed with binary logistic regression.**

|  |  | Prevalence of mental health problem % | | | |
|---|---|---|---|---|---|
| **Predictors** |  | **Male** | | **Female** | |
|  | **Categories** | **UAOR(95% CI)** | **AOR(95% CI)** | **UAOR(95% CI)** | **AOR(95% CI)** |
| Age group (yrs) | 11–13 | Ref | Ref | Ref | Ref |
|  | 14–16 | 1.27(0.51–3.18) | 0.959(0.25–3.7) | 0.98 (0.44–2.16) | 0.65 (0.20–2.06) |
|  | 17–19 | 1.16(0.48–2.84) | 0.643(0.06–6.28) | 1.20 (0.55–2.64) | 0.53 (0.08–3.38) |
| Grade level | Upper elementary (5–8) | 1.08(0.51–2.31) | 1.44(0.50–4.15) | 1.57 (0.70–3.50) | 2.60 (0.95–7.10) |
|  | Lower Secondary (9–10) | 1.02(0.48–2.18) | 1.11(0.50–2.46) | 2.22 (1.00–4.92)* | 2.73 (1.19–6.29)* |
|  | Upper secondary (11–12) | Ref | Ref | Ref | Ref |
| Experience on psychoactive substances use |  |  |  |  |  |
| Self-experience | No | Ref | Ref | Ref | Ref |
|  | Yes | 0.45(0.22–0.93* | 2.20(1.07–4.52)* | 2.17 (0.95–4.95) | 2.16 (0.95–4.92) |
| Any family experience | No | Ref | Ref | Ref | Ref |
|  | Yes | 2.01(1.01–4.00)* | 2.04(1.01–4.09)* | 1.07(0.63–1.84) | 1.06 (0.62–1.81) |
| Parents practicing corporal punishment | No | Ref | Ref | Ref | Ref |
|  | Yes | 1.24 (0.69–2.24) | 1.24 (0.69–2.23) | 1.43 (0.82–2.51) | 1.33 (0.75–2.36) |
| Adolescents report perceived worry about family problem | No | Ref | Ref | Ref | Ref |
|  | Yes | 1.186(0.64–2.21) | 1.17(0.62–2.20) | 1.24 (0.73–2.12) | 1.20 (0.70–2.06) |
| Mother education level | Non-educated | Ref | Ref | Ref | Ref |
|  | Elementary | 1.31(0.50–3.43) | 1.30 (0.49–3.41) | 1.99 (0.57–6.96) | 0.73 (0.29–1.87) |
|  | Secondary | 1.30(0.66–2.57) | 1.30 (0.66–2.58) | 1.50 (0.34–6.70) | 1.37 (0.78–2.42) |
|  | College or above | 0.63(0.18–2.20) | 0.64 (0.18–2.24) | 2.71 (0.74–9.93) | 0.52 (0.15–1.81) |
| Father education level | Non-educated | Ref | Ref | Ref | Ref |
|  | Elementary | 1.51 (0.47–4.82) | 1.48 (0.46–4.74) | 1.03 (0.21–5.14) | 1.00 (0.20–4.99) |
|  | Secondary | 1.60 (0.83–3.09) | 1.61 (0.83–3.11) | 1.04 (0.58–1.84) | 1.07 (0.60–1.90) |
|  | college or above | 1.42 (0.64–3.16) | 1.45 (0.65–3.23) | 1.23 (0.61–2.46) | 1.20 (0.60–2.41) |
| Mother job | House wife/Unemployed | Ref | Ref | Ref | Ref |
|  | Work in private | 0.79 (0.41–1.51) | 0.79 (0.41–1.51) | 1.40 (0.77–2.53) | 1.40 (0.77–2.54) |
|  | Employed | 0.44 (0.16–1.22) | 0.44 (0.16–1.21) | 0.55 (0.22–1.42) | 0.55 (0.21–1.44) |
| Father job | Unemployed/Jobless | Ref | Ref | Ref | Ref |
|  | Work in private | 1.24 (0.66–2.32) | 1.24 (0.66–2.33) | 0.75(0.43–1.33) | 0.77(0.44–1.36) |
|  | Employed | 2.56 (0.75–8.74) | 2.52 (0.73–8.70) | ns | ns |

\*\* Significant at p<0.01

\* Significant at p<0.05

From Table 5, the analysis found strong negative correlations between total difficulties and mental well-being, mental health literacy, and various problem behaviours. Additionally, there were moderate negative correlations between mental well-being and conduct, hyperactivity, and peer problems. Overall, the findings suggest that higher levels of total difficulties are associated with poorer mental health outcomes and increased problem behaviours.

## 4. Discussion

This study aimed to investigate the prevalence of mental health problems among adolescents in Dire Dawa, Ethiopia, and identify associated factors. Our findings reveal a concerning prevalence of mental health issues, with rates ranging from 15.9% to 28.8%. Internalizing problems, emotional problems, and peer relationship problems were the most common issues reported.

**Table 5. Correlations among mental health variables.**

| Variables | 1 | 2 | 3 | 4 | 5 | 6 | 7 | 8 | 9 |
|---|---|---|---|---|---|---|---|---|---|
| 1. SDQ Total difficulties problems score | 1 | -.319** | -.124** | .652** | .700** | .749** | 617** | .825** | .863** |
| 2. Mental wellbeing index | -.319** | 1 | .160** | -.140** | -.281** | -.293** | -.130** | -.258** | -.279** |
| 3. MHL Total mean score | -.124** | .160** | 1 | -.135** | -.136** | -0.006 | -.088* | -.166** | -0.051 |
| 4. SDQ Conduct problems subscale mean score | .652** | -.140** | -.135** | 1 | .345** | .228** | .294** | .812** | .319** |
| 5. SDQ Hyperactivity problems subscale mean score | .700** | -.281** | -.136** | -.136** | 1 | .385** | .197** | .828** | .380** |
| 6. SDQ Emotional problems subscale mean score | .749** | -.293** | -0.006 | .228** | .385** | 1 | .272** | .376** | .862** |
| 7. SDQ Peer problems subscale mean score | 617** | -.130** | -.088* | .294** | .197** | .272** | 1 | .298** | .721** |
| 8. SDQ Externalizing Problems subscale mean score | .825** | -.258** | -.166** | .812** | .828** | .376** | .298** | 1 | .427** |
| 9. SDQ Internalizing problems subscale mean score | .863** | -.279** | -0.051 | .319** | .380** | .862** | .721** | .427** | 1 |

** Correlation is significant at the 0.01 level (2-tailed)

* Correlation is significant at the 0.05 level (2-tailed)

Adolescents with personal or family histories of psychoactive substance use exhibited significantly higher rates of mental health problems. Female adolescents from upper elementary and lower secondary grades were also at increased risk. While age, parental education, and employment status did not significantly impact mental health, these results underscore the need for targeted interventions to address the mental health needs of vulnerable adolescents.

The Ethiopian government has taken several interventions to mitigate the adolescent mental health issue, establishing mental health facilities nationwide [66]. Despite the delayed initiation of the policy and lack of established systems and practices, there has been growing recognition of the importance of adolescent mental health promotion. The Ethiopian government has shown commitment to promoting mental health and addressing mental health issues in schools, for instance, by employing trained school counselors and establishing clubs in secondary schools. The government has also trained health professionals in mental health care who are better equipped to identify and treat mental health problems in adolescents—implementing psycho-education programs in schools, public awareness campaigns, and training for community leaders and media outlets for adolescents, among others [4]. The state mechanism in promoting mental health literacy in Ethiopia is that the Ministry of Health is responsible for developing and implementing mental health policies and programs and provides training for health professionals in mental health care [4]. Stakeholders including professional associations, such as the Ethiopian Mental Health Association and the Ethiopian Psychiatric Association, are involved in policy consultation, provide training and support for health professionals, and promote mental health research in Ethiopia. These policy and programs are cascaded along the government administrative levels for actual implementation. However, there is still a paucity of evidence about the mental health status of adolescents and factors that affect mental health problems and factors that contribute for effective interventions to improve adolescents' mental health.

As a screening tool for mental health problems in children and adolescents, the SDQ is a valuable instrument [67]. However, it is important to note that the cut-off scores for the SDQ can vary across different populations [68]. In our study, we found that the cut-off scores for total difficulties and hyperactivity were slightly lower than the baseline, while the cut-off scores for internalizing were relatively higher (**Table 2**). These variations in cut-off scores can be attributed to cultural differences and other contextual factors [68]. Using locally derived cut-off scores ensures the cultural equivalence and applicability of the SDQ to the specific population being studied. While the SDQ is a useful screening tool, it is not a definitive diagnostic

instrument [67]. If a child scores above the cut-off score, it indicates an increased risk of mental health problems but does not necessarily confirm a diagnosis. Further assessment is necessary to make a definitive diagnosis [43, 46]. These findings implied that the adolescent population in urban Ethiopia is more disadvantaged than the population on which the original cut-off scores were based. Adolescents in urban Ethiopia have been exposed to stressors like poverty, violence, or trauma. This adolescent population has less access to mental health services and is more likely to underreport mental health problems.

As shown in **Fig 1,** the present finding was within the same range of the Ethiopian national level prevalence of adolescent mental health problems, reportedly ranging from 17–23%, like other study results in the Ethiopian context [69–71]. In our study, prevalence of total difficulties (14.0–24.5%), internalising (14.9–28.8%), emotional (10.4–25.5%), and peer relationship (17.8–25.5%) was similar that was within the range of global prevalence (10–20%) [72]. Several studies showed discrepancies in adolescent mental health problems prevalence from region to region and country to country [73–76]. Similarly, these differences exist across socio-demographic characteristics [77]. From the present study, mental health problems expressed by the total strength difficulties and its subscales of problems were higher among female adolescents than male adolescents. A study among Indian children, the association between the female gender and total strength difficulties score was significant [73]. Being female was associated with emotional problems [73]. A cross-national study across 73 countries on the gender gap in adolescent mental health showed that girls have worse average mental health than boys [78]. The possible explanations include but are not limited to socio-cultural factors, gender stereotypes, trauma, and abuse that disproportionally affect girls [22, 79]. In contrast, conduct and hyperactivity/inattention problems were significantly more severe among males than females [65]. Consistent with other studies, the prevalence of conduct and externalizing problems was more significant among male adolescents than females. For example, a study in Northeast china showed a similar result [77]. A study among children and adolescents from seven European countries (Italy, Netherlands, Germany, Romania, Bulgaria, Lithuania, and Turkey) reported that externalized problems were consistently higher in males than females and reversed for internalized problems [74]. One possible explanation might be that mental health status is shaped by social, economic, and physical environments leading to inequalities that disproportionally affect gender differences and are heavily associated with risk factors for many common mental disorders [75]. According to the WHO, the relationship between the prevalence of mental health problems and poverty indices was statistically significant [76]. These poverty indices are education disparity, low income, a lack of material goods, job, and housing obstacles. For several cultural and traditional reasons, these factors impact females more than boys [76].

Odds of mental health problems were significantly associated with some socio-demographic characteristics (p<0.05) differing in magnitude across these characteristics. In this study, the prevalence of mental health problems was significantly higher for females in upper elementary and lower secondary levels than for upper secondary grade levels (p<0.05), consistent with reports of several studies [75, 76]. According to WHO report, educational status is one of the major factors determining adolescent mental health outcomes [76]. However, the difference was insignificant for male adolescents in this study. Furthermore, differences in prevalence have existed but little across the three age groups(early, middle and late adolescents) maternal and paternal education level, and job types or level of employment (p>0.05). The present study's prevalence of mental health problems was twice high among adolescents with a self and/or family members' experience of psychoactive substance use (p<0.05). On the contrary, a study in Northeast China showed that the prevalence of any mental disorders and internalizing disorders was significantly lower in younger adolescents than in elders. Children

aged 11–14 years had the highest prevalence of internalizing disorders [77]. It was also inconsistent with a study in developing countries showing that lower maternal education was significantly associated with abnormal total SDQ scores [31].

The present study revealed mental health literacy was negatively correlated with strength difficulties scores and positively correlated with mental well-being; both were significantly correlated but weaker in magnitude. These findings were consistent with existing theories and the results of previous studies. For instance, a study demonstrated that an inadequate level of mental health literacy was negatively associated with depression [39]. The prevalence of depression was 1.52 times higher in those with levels of inadequate MHL than in those who demonstrated an adequate level of MHL [39]. Another study depicted MHL as an explanatory variable of the mental well-being of adolescents, which has a positive correlation with it [80]. A possible explanation for the negative association between mental health literacy and strength difficulties scores might be that low mental health literacy contributes to the mental health promotion and prevention gap [31, 34, 81, 82]. Low mental health literacy exacerbates barriers and difficulties in accessing treatment for mental health problems [24, 83].

Mental health literacy significantly influences adolescents' perceptions and emotional responses [80], reflecting mental health status and subjective mental well-being. Several studies revealed that mental health literacy had a significant relationship with mental health status [39, 84, 85]. A higher level of mental well-being has been reported among adolescents with better know-how about obtaining mental health services [80]. Adolescents' awareness of symptoms, causes, and mental illness treatment contributes to favorable attitudes toward seeking help [15]. Similarly, adolescents with better mental health literacy are less likely to engage in problematic health behaviours and have better help-seeking intention and self-efficacy, contributing to a better mental health and well-being [86]. Mental health literacy is about the information, attitudes, and awareness of mental health concerns that improve self-efficacy, help-seeking intention, and health behaviour, influencing long-term health outcomes and quality of life. A lack of understanding of the symptoms and nature of mental health disorders and alternate sources of support could explain teenagers' low help-seeking intention and behaviour [15]. Of course, help-seeking intention and self-efficacy may reflect only one of several mechanisms. Better mental health literacy leads to better mental well-being. Higher help-seeking intention and self-efficacy would only partially mediate the relation between higher mental health literacy and better mental well-being. Components of health behavior expressed in actions, targets, contexts, and time are the most determining factors of mental health outcomes [87, 88].

Mental health literacy influences mental health services and help-seeking for treatment [89]. Hence, mental health literacy is essential to improve but not always corresponds to help-seeking behaviours [90]. Jorm and his colleagues revealed that changing knowledge of mental health in principle is vital; however, changing heartily emotional reactions to practice caring and preventing mental disorders may be much more challenging [10]. Help-seeking, interchangeably health-seeking, has become one of the essential perspectives in understanding causes and factors for patient delay from appropriate action across different health conditions [91]. Despite the prevalent mental health problems and poor well-being, adolescents' unwillingness and low intention to seek help usually result in delays in timely treatment [32]. Most adolescents with mental health problems are reluctant and miss the use of healthcare for mental health, which increases the complexity of social, mental, and general health outcomes [92]. The findings from this study implied that promoting mental health literacy can improve subjective mental well-being [80].

The recently revised policy released by The Ethiopian Ministry of Health in 2017 aimed to improve adolescents' mental health by providing early identification, prevention, and

treatment services [4, 21]. Following the policy, programs were launched in 2018, including school-based mental health programs, community-based mental health services, and training for health care providers [21]. Empirical evidence on early identification of vulnerable groups, individuals reporting symptoms of mental health difficulties and the associated actors could enhance the impact and effectiveness of these programs implementation [4, 21]. In this regard, demographic, socioeconomic and cultural appropriate approaches should be applied to mental health policies for mental health promotion and mental illness prevention.

## 5. Limitations

The study has some inherent limitations. It is not applicable to generalize these findings across the country; because Ethiopia exhibits multicultural and multiethnic diversity and multiple socio-demographic backgrounds. It could not even be generalized to all adolescents of the study area, because the samples were selected from the age group (11–19) and grade level (5–12), adolescents at the age of 10 years and lower grade levels were excluded during the first cycle of elementary schools during school sampling. The present study is conducted on the urban adolescents and did not address rural adolescents. The study was quantitative and lacked the qualitative aspect of how and why these findings were observed.

## 6. Conclusion

This study found a concerning prevalence of mental health problems among adolescents in Dire Dawa, Ethiopia, aligning with global estimates. These problems included total difficulties, internalizing problems, emotional problems, and peer relationship problems. Adolescents with personal or family histories of substance use were at significantly higher risk. Females and younger adolescents were also more likely to experience mental health issues. Self-psychoactive substance uses and family members experiencing psychoactive substance use were associated with increased mental health difficulties. Female adolescents were more likely to experience mental health problems than males, as indicated by higher scores on total difficulties and its subscales. This aligns with previous research.

Mental health literacy is negatively correlated with total difficulties and positively associated with mental well-being and the study highlights the importance of improving mental health literacy among adolescents in Dire Dawa. These findings support existing theories and suggest that promoting mental health literacy can improve adolescent mental health. Prioritizing vulnerable groups and individuals experiencing mental health difficulties is crucial. This can be achieved through targeted interventions that consider gender, age, family background, and other socio-demographic factors. Future research should explore these factors in more depth using qualitative methods and should explore additional factors influencing mental health. Future studies with larger sample sizes should conduct multivariable analyses to further explore the complex relationships between mental health literacy, well-being, and mental health problems while controlling for potential confounding factors.

## Supporting information

**S1 Data.**
(SAV)

## Acknowledgments

This study is a part of doctoral thesis of the first author Hailemariam Mamo Hassen under the supervision of Prof. Manas Ranjan Behera at the School of Public Health, Kalinga Institute of

Industrial Technology (KIIT) University, Bhubaneswar, India. The authors would like to thank the survey respondents for their participation and contributions to the study.

## Author Contributions

**Conceptualization:** Hailemariam Mamo Hassen.

**Formal analysis:** Hailemariam Mamo Hassen.

**Investigation:** Hailemariam Mamo Hassen.

**Methodology:** Hailemariam Mamo Hassen, Manas Ranjan Behera.

**Supervision:** Manas Ranjan Behera.

**Validation:** Deepanjali Behera, Ranjit Kumar Dehury.

**Writing – original draft:** Hailemariam Mamo Hassen.

**Writing – review & editing:** Hailemariam Mamo Hassen, Manas Ranjan Behera, Deepanjali Behera, Ranjit Kumar Dehury.

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
