## [Decision Letter · Decision Letter 0]

16 Aug 2023

PONE-D-23-13300Mental Health Issues and the Association of Mental Health Literacy among Adolescents in Urban EthiopiaPLOS ONE

Dear Dr. Hassen,

Thank you for submitting your manuscript to PLOS ONE. After careful consideration, we feel that it has merit but does not fully meet PLOS ONE’s publication criteria as it currently stands. Therefore, we invite you to submit a revised version of the manuscript that addresses the points raised during the review process.

We look forward to receiving your revised manuscript.

Kind regards,

I. Marion Sumari-de Boer, Ph.D

Academic Editor

PLOS ONE

Journal Requirements:

3. Please ensure that you refer to Figure 1 in your text as, if accepted, production will need this reference to link the reader to the figure.

4. Please include your tables as part of your main manuscript and remove the individual files. Please note that supplementary tables (should remain/ be uploaded) as separate "supporting information" files.

Reviewers' comments:

Reviewer's Responses to Questions

**Comments to the Author**

1. Is the manuscript technically sound, and do the data support the conclusions?

Reviewer #1: Yes

Reviewer #2: Yes

2. Has the statistical analysis been performed appropriately and rigorously? 

Reviewer #1: Yes

Reviewer #2: Yes

3. Have the authors made all data underlying the findings in their manuscript fully available?

Reviewer #1: Yes

Reviewer #2: Yes

4. Is the manuscript presented in an intelligible fashion and written in standard English?

Reviewer #1: Yes

Reviewer #2: Yes

5. Review Comments to the Author

Reviewer #1: The article is quite an important work which can be a value addition to the literature on mental health especially in Ethiopian region. However, the article needs minor revision before being accepted for the publication. The comments are as below,

1) The precise location of the focus of the study is better to be mentioned in the abstract to main transparent claims and ground of study.

2) The article needs justification of the choice of themes and methods used. For example, it is always better to connect adolescents'' mental health with life span development and justify the argument in such line of thought. Similarly, more justification is needed in choice of SDQ, WHO-5 to strengthen the method section.

3) In discussion section, the claim "the cut-off score ... for internalizing scores" needs expanded rationalisation and justification to make it more relevant for the purpose of the article.

4) Page 8, third paragraph last sentence - " A cross sectional study across 73 countries..."; p.11 conclusion statement "Female adolescents ..." need strengthened rationalisation for what it implies for this study.

5) More focus is needed in following the prescribed reference style accurately and consistently, both for in text citation (P.10) and reference list (Ref 4,5, 12,16).

Overall, its a good work to be accepted for publication with these minor revisions.

Reviewer #2: The introduction section provides an exhaustive literature review on mental health issues. When explaining a scenario, start with the global scenario followed by Africa, Ethiopia, and then the situation of mental health literacy in the study area.

What is the government intervention to mitigate the issue? What are the state mechanisms promoting mental health literacy in Ethiopia? The author can discuss these interventions in discussion sections.

The author should discuss their policy and programme related to mental health for a more comprehensive understanding.

The author should highlight the key inference of the present study, how this study enhances the knowledge of mental health literacy among young adolescents in Ethiopia.

Suggest a few major findings, how the demographic and socioeconomic factors are leading in addressing the current mental health literacy issues in Ethiopia. Is it applicable across Ethiopia, irrespective of all ethnic groups?

6. PLOS authors have the option to publish the peer review history of their article (what does this mean?). If published, this will include your full peer review and any attached files.

Reviewer #1: **Yes: **Rhyddhi Chakraborty Ph.D., FHEA, FRSPH

Reviewer #2: No

---

## [Author Response · Author response to Decision Letter 0]

9 Nov 2023

Response to Reviewers

Title: Mental health issues and the association of mental health literacy among adolescents in urban Ethiopia (PONE-D-23-13300)

Dear Editor, 

We thank you for your and the reviewers' constructive feedback and the opportunity to revise and resubmit our manuscript. We carefully considered the comments, and the manuscript was revised. We appreciate and thank you very much for your feedback on the suggestion. We followed PLOS ONE's style requirements and moved the ethics statement into the Methods section of the manuscript. We referred to Figure 1 and the tables in the text as part of the main manuscript. We reviewed the reference list to ensure it is complete and correct to check for errors. We have addressed the feedback accordingly. 

Three separate papers are submitted to the system: one has a marked-up copy of the manuscript that highlights changes with track changes, an unmarked version of the revised paper without tracked changes, and a rebuttal letter for each point raised by the academic Editor and reviewers. We are confident that the updated version is appropriate for publishing and anticipate hearing from you soon. 

With regards,

Hailemariam Mamo Hassen (PhD)

On behalf of the authors.

Point-by-point response for Reviewers' comments

Dear reviewers, we are grateful for the feedback and constructive comments helpful for improving the manuscript quality. 

Reviewer #1 Comments

Reviewer comment: The article is quite an important work that can be a valuable addition to the literature on mental health, especially in the Ethiopian region. However, the article needs minor revision before being accepted for publication. The comments are addressed below. 

Author's response: Thank you. We appreciate your constructive comments and agree on that, and the comments are addressed below. 

Reviewer comment: The precise location of the focus of the study is better mentioned in the abstract to maintain transparent claims and ground of study.

Author's response: Thank you very much, dear reviewer. We mentioned the precise location of the focus of the study in the abstract on page 2, line #1, that the study was conducted in Dire Dawa City, Eastern Ethiopia. The details are also already presented in the main body of the paper. 

Reviewer comment: The article needs justification for the themes and methods used. For example, it is always better to connect adolescents'' mental health with life span development and justify the argument in such a line of thought. 

Authors' response: Thank you for your comment. We have added more justifications for the choice of themes and methods used to connect adolescents" mental health with life span development. You may kindly refer to the revised manuscript (Page 4) 

Reviewer comment: Similarly, more justification is needed in the choice of SDQ and WHO-5 to strengthen the method section.

Authors' response: Once again, we appreciate your concern. We justified the choice of SDQ, WHO-5, for the measurement of mental health status in the method section. You may kindly refer to the revised manuscript (Page 5) 

Reviewer comment: In the discussion section, the claim "the cut-off score ... for internalizing scores" needs expanded rationalization and justification to make it more relevant to the article's purpose.

Authors' response: Thank you, dear reviewer. We explained the claim and rationalization/justification on "the cut-off score ... for internalizing scores" and other subscales in the discussion section; you may kindly refer to the revised manuscript (Page 23) 

Reviewer comment: Page 8, third paragraph last sentence - "A cross-sectional study across 73 countries..." p.11 conclusion statement "Female adolescents ..." need to strengthen rationalization for what it implies for this study.

Authors' response: Thank you so much for your important concern. We Strengthened rationalization for what it implies for this study for the sentences on: 

• On the revised manuscript, page 24, fourth paragraph - "A cross-sectional study across 73 countries..."(comment incorporated on page line in the revised version. You may kindly refer to the revised manuscript (Page 24) 

• On the revised manuscript page 29, second paragraph conclusion statement You may kindly refer to the revised manuscript (Page 29) 

Reviewer comment: More focus is needed on following the prescribed reference style accurately and consistently for in-text citation (P.10) and reference lists (Ref 4,5, 12,16).

Authors' response: We appreciate your concern. We checked and revised all the references, including these commented references, as per the PLOS ONE reference style for its accuracy and consistency, including the commented references in-text citation (P.10) and reference list (Ref 4,5, 12,16) 

Reviewer comment: Overall, it is a good work to be accepted for publication with these minor revisions.

Authors' response: Thank you very much.

Reviewer #2 Comments:

Authors' response to Reviewer #2 Comments:

Dear reviewer #2, we acknowledge all insightful comments and fully agree with the suggestions. We addressed the comments one by one.

Reviewer comment: The introduction provides an exhaustive literature review on mental health issues. When explaining a scenario, start with the global scenario followed by Africa, Ethiopia, and mental health literacy in the study area.

Authors' response: Thank you very much, dear reviewer. We added more explanations, starting with the global scenario followed by Africa, Ethiopia, and mental health literacy in the study area. You may kindly refer to the revised manuscript (Introduction Page 3, first paragraph) 

Reviewer comment: What is the government intervention to mitigate the issue? What are the state mechanisms promoting mental health literacy in Ethiopia? The author can discuss these interventions in discussion sections.

Authors' response: Thank you so much for your essential concern. We discussed the government intervention to mitigate the issue, state mechanisms promoting mental health literacy in Ethiopia, and their policy and programme related to mental health for a more comprehensive understanding in the discussion sections with additional references to the national health policy and programme. You may kindly refer to the revised manuscript ( throughout the discussion and mainly in the first paragraph of the discussion part, Page 22) 

Reviewer comment: The author should discuss their policy and programme related to mental health for a more comprehensive understanding.

Authors' response: We appreciate your crucial concern. We discuss the policy and programme related to mental health in Ethiopia throughout the manuscript for a more comprehensive understanding and, most importantly, discussion in the last paragraph of the discussion. (Pages 27-28)

Reviewer comment: The author should highlight the key inference of the present study, how this study enhances the knowledge of mental health literacy among young adolescents in Ethiopia.

Authors' response: Thank you so much. We revised and highlighted the key inference of the study, noting the limitations of generalization in the limitation and conclusion parts (pages 28&29). 

Reviewer comment: Suggest a few major findings, how the demographic and socioeconomic factors are leading in addressing the current mental health literacy issues in Ethiopia.

Authors' response: Thank you, dear reviewer. We revised the manuscript, presenting the significant findings and how the demographic and socioeconomic factors imply addressing the current mental health literacy (Page 29 paragraph 2)

Reviewer comment: Is it applicable across Ethiopia, irrespective of all ethnic groups? 

Authors' response: We appreciate your crucial concern. As it is presented in the limitation section, It is not applicable to generalize these findings across the country because Ethiopia exhibits multicultural and multiethnic diversity and multiple socio-demographic backgrounds. It could not even be generalized to all adolescents of the study area. (Page 28, limitation, paragraph 1)

---

## [Decision Letter · Decision Letter 1]

25 Jan 2024

PONE-D-23-13300R1Mental health issues and the association of mental health literacy among adolescents in urban EthiopiaPLOS ONE

Dear Dr. Hassen,

Thank you for submitting your manuscript to PLOS ONE. After careful consideration, we feel that it has merit but does not fully meet PLOS ONE’s publication criteria as it currently stands. Therefore, we invite you to submit a revised version of the manuscript that addresses the points raised during the review process.

We look forward to receiving your revised manuscript.

Kind regards,

Marianne Clemence, Staff Editor, on behalf of

Tesera Bitew, PhD

Academic Editor

PLOS ONE

Journal Requirements:

**Ethical requirements - please note that if you do not address these requests your manuscript may be rejected**

1) You indicated that you had ethical approval for your study. In your Methods section, please ensure you have also stated whether you obtained consent from parents or guardians of the minors included in the study or whether the research ethics committee or IRB specifically waived the need for their consent.

2) We note that your study was approved by both the ethics committees of Haramaya University and KIIT University, but you have only provided a copy of the approval received from Haramaya University. Please upload a copy (file type Other) of the original ethics approval letter received from KIIT University, as required by question 1 of the human subject research checklist. If the original letter is not in English, please also provide an English translated version in the supporting file. 

3) Whilst reviewing the ethics document issued by Haramaya University, we noted that the signature from the Chairperson is dated 11/02/2018 whilst the Secretary is dated 11/02/2020. Could you please explain this discrepancy. If a correction has been issued by the ethical review board, could you please supply a copy as file type Other.

Additional Editor Comments:

Dear author,

It is my pleasure that your manuscript has been accepted once you accomodate comments raised by the two reviewers.

reviewer 1

reviewer 2:

Would you please review your claims on the discussion section regarding government actions taken at school level with regard to mental health promotion and prevention as there are no promising activities? These are actually policy promises than actual practices.

I think you better consider including the predictor variables in the method part.

Your sampling strategy should discuss the clusters selected all the stages precisely.

The statement on page 13 that begins with "These mental health problems were..." does not provide a clear comparison.

Would you please mention any bases for your age categorization and its subsequent implication for mental health promotion and prevention?

Reviewers' comments:

Reviewer's Responses to Questions

**Comments to the Author**

1. If the authors have adequately addressed your comments raised in a previous round of review and you feel that this manuscript is now acceptable for publication, you may indicate that here to bypass the “Comments to the Author” section, enter your conflict of interest statement in the “Confidential to Editor” section, and submit your "Accept" recommendation.

Reviewer #2: All comments have been addressed

Reviewer #3: (No Response)

2. Is the manuscript technically sound, and do the data support the conclusions?

Reviewer #2: Yes

Reviewer #3: Partly

3. Has the statistical analysis been performed appropriately and rigorously? 

Reviewer #2: Yes

Reviewer #3: Yes

4. Have the authors made all data underlying the findings in their manuscript fully available?

Reviewer #2: Yes

Reviewer #3: Yes

5. Is the manuscript presented in an intelligible fashion and written in standard English?

Reviewer #2: Yes

Reviewer #3: Yes

6. Review Comments to the Author

Reviewer #2: Author has addressed all the comments and suggestions. Mental health literacy is an emerging issues which demand more attention for overall health and well-being.

Reviewer #3: Would you please review your claims on the discussion section regarding government actions taken at school level with regard to mental health promotion and prevention as there are no promising activities? These are actually policy promises than actual practices.

I think you better consider including the predictor variables in the method part.

Your sampling strategy should discuss the clusters selected all the stages precisely.

The statement on page 13 that begins with "These mental health problems were..." does not provide a clear comparison.

Would you please mention any bases for your age categorization and its subsequent implication for mental health promotion and prevention?

7. PLOS authors have the option to publish the peer review history of their article (what does this mean?). If published, this will include your full peer review and any attached files.

Reviewer #2: No

Reviewer #3: No

---

## [Author Response · Author response to Decision Letter 1]

12 Feb 2024

Response to Reviewers

Title: Mental health issues and the association of mental health literacy among adolescents in urban Ethiopia (PONE-D-23-13300)

Dear Editor, 

We thank you for your and the reviewers' constructive feedback and the opportunity to revise and resubmit our manuscript. We carefully considered each point raised by the academic editor and reviewers and revised the manuscript. We appreciate and thank you very much for the feedbacks and comments. We followed PLOS ONE's style requirements. 

Point raised by the academic editor were regarding whether consent obtained from parents or guardians of the minors, ethics approval letter from two institutions of which one was missing and discrepancy in date difference for secretary and chair person for ethics approval letter obtained from Haramaya University and concern to ensure completeness and correctness of references. 

Dear Academic editor, regarding consent from parents or guardians of the minors, written informed consent was obtained from parents of adolescents aged under fifteen years had offered their consent and adolescents of fifteen and above years were factored out following the school counsellors and school principals adequate explanation about their level of maturity and decisional capacity ((page 10) with due reference of the guideline International Ethical Guidelines for Health-related Research Involving Humans: Prepared by the Council for International Organizations of Medical Sciences (CIOMS) in collaboration with the World Health Organization (WHO,2016). However, respective school principals as a duly authorized representative had expressed their consent both orally and signed on consent forms for all participants. 

Regarding the ethics approval letters, ethics approval letter from the two institutions (KIIT University and Haramaya University) are merged and attached in this revised submission. The discrepancy of the date of signature from the Chairperson (dated 11/02/2018) and Secretary (dated 11/02/2020) on same document was confirmed as typo while the typist was writing and the letter issued unnoticed. We understand that it requires a correction. However, our application for correction from the Institutional review board of Haramaya University failed because the Institutional review board has restructured and the Signers were changed by other individuals. We sincerely request your esteemed editor to understand that this discrepancy is typo while the typist was writing and we kindly request you recognize this issue. 

We reviewed the reference lists to ensure it is complete and correct. 

Dear editor, we appreciate the comments raised by the reviewers and we accommodate each comment one by one in the manuscript indicated by track change. 

Three separate papers are submitted to the system: one has a marked-up copy of the manuscript that highlights changes with track changes, an unmarked version of the revised paper without tracked changes, and a rebuttal letter for each point raised by the academic Editor and reviewers. We are confident that the updated version is appropriate for publishing and anticipate hearing from you soon. 

With regards,

Hailemariam Mamo Hassen (PhD), (First and corresponding author)

On behalf of the authors.

Point-by-point response for Reviewers' comments

Dear reviewers, we are grateful for the feedback and constructive comments helpful for improving the manuscript quality. 

Reviewer #2 Comments

Thank you very much, dear reviewers. 

Reviewer #2 commented that we authors have addressed all the comments and suggestions stating that mental health literacy is an emerging issue which demand more attention for overall health and well-being.

Authors' response Thank you very much: 

Reviewer #3 Comments

Author's response: Thank you. We appreciate your constructive comments and agree on the comments and addressed accordingly. . 

Reviewer #3 comment: Would you please review your claims on the discussion section regarding government actions taken at school level with regard to mental health promotion and prevention as there are no promising activities? These are actually policy promises than actual practices.

Authors' response: Dear Reviewer, thank you so much for your essential concern. We agree that the government actions taken at school level with regard to mental health promotion and prevention are policy promises than actual practices. Accordingly, we discussed these points. Despite the delayed initiation of the policy and lack of established systems and practices, there has been growing recognition of the importance of adolescent mental health promotion. The Ethiopian government has shown commitment to promoting mental health and addressing mental health issues in schools, for instance, by employing trained school counselors and establishing clubs in schools in secondary schools. (stated on page 25) 

Reviewer #3 comment: I think you better consider including the predictor variables in the method part.

Author's response: Thank you very much, dear reviewer. We included the predictor variables briefly in the method part (page 8). “The predictor variables mainly focusing on age, grade level, self or any family member experience with psychoactive substance use, parents practicing corporal punishment, perceived worry about family problems, parents education, and job status were collected by using questionnaires.” 

Reviewer #3 comment: Your sampling strategy should discuss the clusters selected all the stages precisely. 

Author's response: Thank you. We mentioned the sampling strategy briefly in the method part (page 8). “A combination of multistage (schools, classrooms, then individual students) and systematic and random (using the list of the students in fixed intervals of their roll numbers) was used to select study participants from public and private schools.”

Reviewer #3 comment: The statement on page 13 that begins with "These mental health problems were..." does not provide a clear comparison. 

Author's response: Once again, we appreciate your concern. We have compared and indicated in the figure 1 in the revised manuscript, page 14. We stated that “These mental health problems were more prevalent for female and male adolescents compared to other age groups in extent of percentage as explicitly visualized in Figure 1 across the three age groups. Prevalence of depression ranged from 18.0%-25.5%, reportedly higher for female adolescents aged 14-16 years ( page 14).

Reviewer #3 comment: Would you please mention any bases for your age categorization and its subsequent implication for mental health promotion and prevention?

Author's response: Thank you very much, dear reviewer. We already mentioned the bases for your age categorization and its subsequent implication on page 11 of the last 4 lines #6-9; “The age range for adolescents and grouping was based on UNICEF adolescents' age categorization as early, middle, and late adolescence age groups, which is helpful to target mental health promotion and prevention efforts during these stages to address their respective mental health demands.”

---

## [Decision Letter · Decision Letter 2]

27 Aug 2024

PONE-D-23-13300R2Mental health issues and the association of mental health literacy among adolescents in urban EthiopiaPLOS ONE

Dear Dr. Hailemariam,

Thank you for submitting your manuscript to PLOS ONE. After careful consideration, we feel that it has merit but does not fully meet PLOS ONE’s publication criteria as it currently stands. Therefore, we invite you to submit a revised version of the manuscript that addresses the points raised during the review process.

We look forward to receiving your revised manuscript.

Kind regards,

Yared Reta Abayneh, MSC in ICCMH

Academic Editor

PLOS ONE

Journal Requirements:

Additional Editor Comments:

Dear Dr. Hailemariam,

I hope you are well. I’m writing regarding your manuscript titled “Mental health issues and the association of mental health literacy among adolescents in urban Ethiopia.” I apologize for the delay in the review process, which was due to difficulties in securing reviewers. We appreciate your patience during this time.

We were able to involve multiple reviewers, whose diverse feedback will strengthen the quality of your manuscript. Based on their comments, we request that you make some minor revisions. Please respond to each point raised in the attached reviews and indicate the changes made in your revised manuscript.

Thank you for your cooperation and understanding. We look forward to receiving your revised submission.

Reviewers' comments:

Reviewer's Responses to Questions

**Comments to the Author**

1. If the authors have adequately addressed your comments raised in a previous round of review and you feel that this manuscript is now acceptable for publication, you may indicate that here to bypass the “Comments to the Author” section, enter your conflict of interest statement in the “Confidential to Editor” section, and submit your "Accept" recommendation.

Reviewer #2: All comments have been addressed

Reviewer #4: All comments have been addressed

Reviewer #5: (No Response)

Reviewer #6: (No Response)

2. Is the manuscript technically sound, and do the data support the conclusions?

Reviewer #2: Yes

Reviewer #4: Partly

Reviewer #5: No

Reviewer #6: Yes

3. Has the statistical analysis been performed appropriately and rigorously? 

Reviewer #2: Yes

Reviewer #4: Yes

Reviewer #5: No

Reviewer #6: Yes

4. Have the authors made all data underlying the findings in their manuscript fully available?

Reviewer #2: Yes

Reviewer #4: Yes

Reviewer #5: (No Response)

Reviewer #6: Yes

5. Is the manuscript presented in an intelligible fashion and written in standard English?

Reviewer #2: Yes

Reviewer #4: Yes

Reviewer #5: No

Reviewer #6: Yes

6. Review Comments to the Author

Reviewer #2: All comments have been address. The literature review is meticulously organised, establishing a strong basis for the research. The technique is thoroughly clarified, ensuring clarity and the capacity to replicate the study. The study is rigorously supported by suitable statistical methodologies, and the findings are clearly presented and explained.

All the best!

Reviewer #4: Thank you author(s) for your valuable research. Because mental health concerns is crucial for policy makers & any concerned bodies of the intended areas especially regarding adolescents. My concerns in general is that; you indicated that the policy makers in the area were you did your research more or less on striving to strengthen the strategy to tackle mental health problems. Thus it is not your concern to mention it in detailed.

Reviewer #5: The authors have undertaken important work and the conclusions could make far reaching contributions to policy. However, I do not believe that adequate rigour has been applied in the analysis and write-up, and the manuscript needs to be edited for grammar and readability. The specific comments will be found in the attached copy of the manuscript. Please find a few key points below:

-Please be specific where you can be. Words like "about" when talking about your own data suggest that you are not sure of the data you are presenting.

- Avoid extraneous details in the discussion. There was too much written about the study tool in the discussion that should be cut or moved to the methods section

- It is curious that a multivariable analysis was not done to test the independence of the observed association between mental health literacy and wellbeing and mental health problems, when several variables that were in your dataset (like age and proxies for socioeconomic status) could easily be confounders. I strongly recommend performing this analysis.

- I recommend that the authors seek independent editing of the manuscript.

Reviewer #6: Thank you for the opportunity to review this work. The manuscript titled under review present important evidence for adolescent mental health in Africa and by extension low-middle income countries. It applied sound analytical approaches to answer the object of the study. However, the manuscript requires major revisions before being accepted for publications.

Kindly find attached my review comments.

Introduction

- Page 5 paragraph 3: The description and justification of the SDQ and WHO wellbeing 5 should be included in the methods section when discussing the instruments used.

- Page 7: Study objective 3 “to examine the impact of socio-demographic characteristics on mental health issues” should be revised as the cross-sectional nature of the study does not allow for cause-and-effect relationships to be established. Also, the analytical framework used could only examine the association between socio-demographic characteristics and health issues.

- Consider revising the introduction, be succinct and thoughts should follow in a logical sequence.

Results

It will be helpful to present a table on the sociodemographic characteristics of the participants before presenting other results. This should indicate the frequencies, percentage, and confidence intervals.

- Page 11 - Table 1 should added to supplementary tables or removed from the manuscript.

- Page 13- the prevalence estimates as stated in the manuscript does not correspond with results in fig 1. E.g. Prevalence of total difficulties ranged from 14.0%-24.5% not “15.9-25.5%”, internalizing 14.9-28.8% not “14.8-28.4%” Please check and revise all the results in the manuscript.

- Report the results in table 2 in the manuscript. Provide a description beneath the table for “a” as denoted in the table.

- “It was twice higher among adolescents with a self and family members experiencing psychoactive substance use (p0.05)” … state the AOR and CI’s.

- Table 3: Use Ref instead of “1” to indicate the reference group in present the results for the logistic regression

- Present the results for table 4 correlational analysis in the manuscript.

The table format is a bit cumbersome. I suggest you review your tables and present them in APA format if not otherwise indicated.

Discussion

While the authors have provided sufficient references to support the findings of the study, the presentation appears not well structured making it difficult to follow.

- Consider beginning your discussion by first providing a summary of your key findings based on the objectives for the study. Then you can proceed to discuss them one after the other in an organized and precise manner.

Conclusions

The conclusion appears a bit elaborate. Please summarize the key objective and key findings and possible recommendation in a precise manner.

Minor revisions

- Page 30 paragraph 3 Kindly revise in-text citation. “Anthony F. Jorm and his colleagues” should read Jorm and his colleagues.

- Kindly review the entire manuscript to ensure that all in-text citations are properly cited.

- The authors should revise the manuscript to improve the flow and readability of the text.

7. PLOS authors have the option to publish the peer review history of their article (what does this mean?). If published, this will include your full peer review and any attached files.

Reviewer #2: No

Reviewer #4: No

Reviewer #5: **Yes: **John-Paul Omuojine, MBChB FWACP(Psych)

Reviewer #6: No

---

## [Author Response · Author response to Decision Letter 2]

25 Sep 2024

A rebuttal letter

Dear Editor and Reviewers,

Thank you for your valuable feedback on our manuscript “Mental health issues and the association of mental health literacy among adolescents in urban Ethiopia (PONE-D-23-13300)”. We are pleased to learn that all reviewers are satisfied with the responses provided to their initial comments. We believe the revised manuscript now fully meets PLOS ONE's publication criteria. For the current version the manuscript, we have carefully considered all of the comments and have made required revisions to address your concerns. All comments have been addressed one by one. The comments raised by the editor and reviewers and authors responses are presented in tabular form for easy reference on the comments and responses. The A marked-up copy of the manuscript with Track Changes is attached for your review.

We believe that the revised manuscript now addresses all of the reviewers' comments and is suitable for publication in PLOS ONE. Thank you again for your time and consideration.

Sincerely, 

Hailemariam Mamo Hassen(Ph.D)

Corresponding author

Comments by the editor and reviewers and Authors response 

Comments Authors responses

Editor comment 

Please review your reference list to ensure that it is complete and correct. If you have cited papers that have been retracted, please include the rationale for doing so in the manuscript text, or remove these references and replace them with relevant current references. Any changes to the reference list should be mentioned in the rebuttal letter that accompanies your revised manuscript. If you need to cite a retracted article, indicate the article’s retracted status in the References list and also include a citation and full reference for the retraction notice. Thank you for your careful review of our manuscript.

We have thoroughly reviewed our reference list to ensure its completeness and accuracy. There are no any retracted references papers.

We request that you make some minor revisions. Please respond to each point raised in the attached reviews and indicate the changes made in your revised manuscript. Thank you for your prompt review of our manuscript. We appreciate the thoughtful comments from the editor and the reviewers and have carefully considered each point raised incorporated the suggested revisions into the revised manuscript attached. Please find below a detailed response to each of the reviewer's comments:

Reviewers Comments 

Reviewer #2 Comments: All comments have been addressed. The literature review is meticulously organized, establishing a strong basis for the research. The technique is thoroughly clarified, ensuring clarity and the capacity to replicate the study. The study is rigorously supported by suitable statistical methodologies, and the findings are clearly presented and explained.

All the best! We would like to express our sincere gratitude. Thank you for your valuable feedback.

Reviewer #4 Comments: Thank you author(s) for your valuable research. Because mental health concerns is crucial for policy makers & any concerned bodies of the intended areas especially regarding adolescents. 

My concern in general is that; you indicated that the policy makers in the area were you did your research more or less on striving to strengthen the strategy to tackle mental health problems. Thus it is not your concern to mention it in detailed. Thank you for your thoughtful comments and appreciation of our research. We acknowledge that mental health concerns are indeed critical for policymakers and relevant stakeholders, particularly among adolescents.

We understand your concern regarding the level of detail provided on the policy makers' efforts to strengthen mental health strategies. 

 We believe that it's important to acknowledge the broader context within which our research is situated. Statements on policy makers and strategies were mentioned regarding a paucity of evidence about the increasing demand for adolescents’ mental health and well-being outcome measures to inform the public and policymakers and professionals involved in policy consultation, and how our study contribute evidences mentioning the delayed initiation of the policy and lack of established systems and practices despite government growing recognition and commitment to promoting mental health. Additionally, we mentioned the existing recently drafted policy released by The Ethiopian Ministry of Health in 2017 aimed to improve adolescents' mental health by providing early identification, prevention, and treatment services. [4,21] and programs launched in 2018, including school-based mental health programs, community-based mental health services, and training for health care providers.[21] 

Therefore, we kindly note that presence of the evidence statement would be essential. 

Thank you. 

Reviewer #5 Comments: 

The authors have undertaken important work and the conclusions could make far reaching contributions to policy. However, I do not believe that adequate rigour has been applied in the analysis and write-up, and the manuscript needs to be edited for grammar and readability. The specific comments will be found in the attached copy of the manuscript. Please find a few key points below: Thank you for your valuable feedback. We have carefully addressed your specific comments and made significant improvements to our manuscript. We've strengthened our methodology, clarified our writing, and refined our conclusions to better reflect our findings and the policy implications. We believe these revisions have enhanced the quality of our work and hope it meets your journal's high standards.

Hence, please find below the detailed responses to each of the comments (a-d). 

a. Reviewer #5 Comments: Please be specific where you can be. Words like "about" when talking about your own data suggest that you are not sure of the data you are presenting. Thank you for your careful review of our manuscript. We appreciate your attention to detail and your suggestion to replace the word "about" where appropriate. We have carefully reviewed our manuscript and have made the necessary changes to provide more precise and accurate information.

We have replaced "about" with specific terms where appropriate (page 2 line27), ensuring that our data is presented clearly and accurately. Additionally, we have provided exact numbers or percentages whenever possible to enhance the precision of our reporting.

We believe that these revisions have strengthened the clarity and accuracy of our manuscript and have addressed your concerns effectively. Thank you again for your valuable feedback.

b. Reviewer #5 Comments: Avoid extraneous details in the discussion. There was too much written about the study tool in the discussion that should be cut or moved to the methods section Thank you very much. Details were presented about the SDQ to the methods section. As the given comment we provided a brief overview in the discussion. We have streamlined the discussion and focused on the key findings and implications of our study removing the extraneous details(page23 line 432-442)

c. Reviewer #5 Comments: It is curious that a multivariable analysis was not done to test the independence of the observed association between mental health literacy and wellbeing and mental health problems, when several variables that were in your dataset (like age and proxies for socioeconomic status) could easily be confounders. I strongly recommend performing this analysis. Thank you for your valuable feedback regarding the inclusion of a multivariable analysis in our study. We understand your recommendation and acknowledge the potential benefits of such an analysis in controlling for confounding variables.

While we agree that a multivariable analysis could provide additional insights into the relationship between mental health literacy, well-being, and mental health problems, we believe that the extensive number of tables required for such an analysis would exceed the limitations imposed by PLOS ONE.

However, we have carefully considered your suggestion and plan to include a multivariable analysis in a future study with a larger sample size. This will allow us to more comprehensively explore the complex relationships between these variables and provide a deeper understanding of the factors influencing adolescent mental health.

We include recommendation that reads “Future studies with larger sample sizes should conduct multivariable analyses to further explore the complex relationships between mental health literacy, well-being, and mental health problems while controlling for potential confounding factors” (Page29 Line 637-640)

We believe that the correlation analysis presented in this study provides a valuable foundation for future research and contributes significantly to our understanding of the topic. Thank you again for your insightful comments. 

d. Reviewer #5 Comments: I recommend that the authors seek independent editing of the manuscript. Thank you for your valuable feedback and recommendation for independent editing. We have carefully considered your suggestion and have taken steps to enhance the clarity and coherence of our manuscript.

In addition to the independent editing conducted by a colleague, all authors have engaged in a thorough review and revision process. We have carefully examined the discussion section and have made necessary adjustments to ensure that our statements are clear, concise, and supported by the evidence presented in the study.

We believe that these combined efforts have significantly improved the overall quality and readability of our manuscript. We are confident that our revised work meets the high standards of your journal.

Thank you again for your time and consideration. We look forward to your further feedback.

Reviewer #6 Comments: 

Thank you for the opportunity to review this work. The manuscript titled under review present important evidence for adolescent mental health in Africa and by extension low-middle income countries. It applied sound analytical approaches to answer the object of the study. However, the manuscript requires revisions before being accepted for publications.

Kindly find attached my review comments. Thank you for your valuable feedback on our manuscript. We appreciate your constructive comments and have carefully considered your comments. We have carefully addressed each of your specific comments and have provided detailed explanations for the changes we have made. Please refer to the revised manuscript for a comprehensive overview of these revisions.

Reviewer #6 Comments on Introduction section 

a. Comment1 Page 5 paragraph 3: The description and justification of the SDQ and WHO wellbeing 5 should be included in the methods section when discussing the instruments used.

 Thank you for your valuable feedback. We have carefully considered your comment regarding the placement of the description and justification of the SDQ and WHO-5 wellbeing index.

In response, we have removed the unnecessary repetition of these details from this section (page 5, lines 98-99) that is already incorporated into the methods section. We believe that this change enhances the clarity and organization of our manuscript.

We appreciate your continued attention to detail and your contributions to improving the quality of our work.

b. Comment2 Page 7: Study objective 3 “to examine the impact of socio-demographic characteristics on mental health issues” should be revised as the cross-sectional nature of the study does not allow for cause-and-effect relationships to be established. Also, the analytical framework used could only examine the association between socio-demographic characteristics and health issues. Thank you for your valuable feedback on our study objective 3. We acknowledge your comment that the cross-sectional nature of our study does not allow for the establishment of cause-and-effect relationships.

In response to your suggestion, we have revised study objective 3 to read as follows: "To examine the association between socio-demographic characteristics and mental health issues." This revised objective more accurately reflects the limitations of our study design and the nature of the analysis we conducted.

We believe that this revision addresses your concerns and provides a more accurate representation of our study's scope and limitations. (Page 7 line 167)

c. Comment3 Consider revising the introduction, be succinct and thoughts should follow in a logical sequence.

 Thank you for your valuable feedback on our introduction. We acknowledge your suggestion to revise it for greater succinctness and logical flow. We have carefully reviewed the introduction and made the necessary adjustments removing unnecessary repetition of these details to ensure that our thoughts are presented in a clear and concise manner. (page 5, lines 98-99)

We believe that these revisions have significantly improved the clarity and coherence of our introduction. Thank you again for your helpful feedback

Reviewer #6 Comments on Results section 

a. Comment1 It will be helpful to present a table on the socio-demographic characteristics of the participants before presenting other results. This should indicate the frequencies, percentage, and confidence intervals Page 11 - Table 1 should added to supplementary tables or removed from the manuscript. Thank you for your valuable feedback regarding the presentation of socio-demographic characteristics in our manuscript. We appreciate your suggestion to provide a more comprehensive overview of our participants' demographics.

In response to your comment, we have added a new Table 1 to the main body of the manuscript, presenting the socio-demographic characteristics of our participants, including frequencies and percentages. We believe that this table will enhance the clarity and understanding of our study's findings.

While we have added Table 1 as you suggested with descriptions (Page 10-11 line 257-286), and we have retained table 1 from previous version. We believe that the information contained in Table 2 (current version) remains crucial for a deeper understanding of our research and its implications providing context and details for the readers. Previous tables number 1,2,3, and 4 became table 2,3,4 and 5 in the current version. We hope that these revisions address your concerns and improve the overall quality of our manuscript. Thank you again for your

b. Comment2 Page 13- the prevalence estimates as stated in the manuscript does not correspond with results in fig 1. E.g. Prevalence of total difficulties ranged from 14.0%-24.5% not “15.9-25.5%”, internalizing 14.9-28.8% not “14.8-28.4%” Please check and revise all the results in the manuscript. Thank you for your careful review of our manuscript. We appreciate your attention to detail and your insightful comments. We have carefully reviewed your comment regarding the discrepancies between the prevalence estimates reported in the text and those depicted in Figure 1. You are absolutely correct. There were errors in the text.

We have thoroughly checked and corrected all the prevalence estimates in the manuscript to accurately reflect the data presented in Figure 1. The revised manuscript now includes the correct prevalence estimates for total difficulties, internalizing difficulties, and other relevant variables. We believe that these changes have strengthened the clarity and accuracy of our findings. (Page 14 line 296, 297& 301)

c. Comment3 Report the results in table 2 in the manuscript. Provide a description beneath the table for “a” as denoted in the table. Thank you for your valuable feedback. We have carefully considered your comments and have revised the manuscript accordingly. Your suggestion to report the results in Table 2 [current version of the manuscript table 3 because of addition of a table] was addressed providing more. (Page 17 line 320-327). We believe this enhancement will improve the clarity and understanding of our findings."

d. Comment4 “It was twice higher among adolescents with a self and family members experiencing psychoactive substance use (p0.05)” … state the AOR and CI’s. Thank you for your careful review of our manuscript. We apologize for any ove

---

## [Editor Report · Decision Letter 3]

9 Oct 2024

Mental health issues and the association of mental health literacy among adolescents in urban Ethiopia

PONE-D-23-13300R3

Dear Dr. Hassen,

We’re pleased to inform you that your manuscript has been judged scientifically suitable for publication and will be formally accepted for publication once it meets all outstanding technical requirements.

Kind regards,

Yared Reta Abayneh, MSC in ICCMH

Academic Editor

PLOS ONE
---

## [Editor Report · Acceptance letter]

15 Oct 2024

PONE-D-23-13300R3 

PLOS ONE

Dear Dr. Hassen, 

I'm pleased to inform you that your manuscript has been deemed suitable for publication in PLOS ONE. Congratulations! Your manuscript is now being handed over to our production team.

Kind regards, 

on behalf of

Dr. Yared Reta Abayneh 

Academic Editor

PLOS ONE